# Impacts of Water Level Fluctuations on Soil Aggregate Stability in the Three Gorges Reservoir, China

**Gratien Nsabimana** [1,2], **Yuhai Bao** [1,*], **Xiubin He** [1], **Jean de Dieu Nambajimana** [1,2], **Mingfeng Wang** [1,2], **Ling Yang** [1,2], **Jinlin Li** [1,2], **Shujuan Zhang** [1,2] and **Dil Khurram** [1,2]

1   Key Laboratory of Mountain Surface Processes and Ecological Regulation, Institute of Mountain Hazards and Environment, Chinese Academy of Sciences, Chengdu 610041, China; gnsabimana1@gmail.com (G.N.); xiubinh@imde.ac.cn (X.H.); njado52@gmail.com (J.d.D.N.); w1129338066@163.com (M.W.); yangling@imde.ac.cn (L.Y.); chinlin_lee@sina.com (J.L.); zsj0816@126.com (S.Z.); dilkhurrm@gmail.com (D.K.)
2   University of Chinese Academy of Sciences, Beijing 100049, China
*   Correspondence: byh@imde.ac.cn; Tel.: +86–28–85020291; Fax: +86–28–85222258

**Abstract:** Aggregate is the basic unit of soil structure, which is crucial to the sustainability of soil system functions such as structural stability and Fertility Maintenance. Three Gorges Dam (TGD) has extensively led to a dramatic hydrological regime alteration, which may consequently affect various soil physical properties. The aim of this study was to investigate the long-run temporal variation of soil aggregate stability as induced by water-level fluctuations in the riparian zone of the Three Gorges Reservoir (TGR). Sampling plots were established along different elevations considering the interval of 5 m, starting from 150 m to 175 m. A Laser Diffraction based analysis that allows the measurement of soil aggregate stability after the removal of soil organic matter helped to particularly study the effect of external factors on soil aggregate stability of the study area. In addition, wet-sieving method considering the effect of chemical binding agents was used to quantify aggregate stability. The present results indicated a significant increase of Mean Volume Diameter, MVD ($p < 0.05$) within the study period. Continuous drying-wetting cycles mended soil aggregate stability with a 14.25% increase of the MVD from 2012 to 2016. In the Water-Level Fluctuation Zone (WLFZ), the lower land has predominantly contributed to the increase of soil aggregate stability compared to upper land, with an increase of 62.19% and 37.81% for MVD, 60.88% and 39.12% for $D_{10}$, 95.34% and 4.66% for $D_{90}$ at lower and upper elevations, respectively. Sediment deposition below 165 m has precluded a direct effect of water stress on soil aggregates, which certainly declined soil disaggregation. The removal of SOM while analyzing aggregate stability by LD may explain the contradiction between the resulted MVD, and the MWD and GMD. The increase of MWD and GMD was mainly attributed to the increase of SOM with $r^2 = 0.89$ ($p < 0.01$) and $r^2 = 0.90$ ($p < 0.01$), while the increase of MVD was highly predicted by the decrease of SOM with $r^2 = 0.88$ ($p < 0.01$). Since this study presents a remarkable change of soil in the riparian area due to dry-wet cycles, our results may help to deeply understand the soil ecology and environmental changes in the WLFZ.

**Keywords:** soil aggregate stability; drying and wetting cycles; water-level fluctuation zone; laser diffraction; three gorges dam

## 1. Introduction

Pedologically and edaphologically, soil structure is referred to as the size, shape, strength, and pore capacity of soil aggregates to sustain and transfer soil fluid materials and the ability to support roots growth and enlargement [1]. Bronick and Lal [2] defined soil aggregates as the basic unit of soil

structure. Broken-down macro-aggregates result in micro-aggregates and primary soil particles which displace and then reorganize into a seal, which eventually led to a decline in infiltration rate and an increase in soil erosion risk [3]. The most important soil aggregate property is its stability to various stresses applied to aggregates. Aggregate stability is primarily considered as the indicator of soil quality. Additionally, it plays an important role in indicating the susceptibility of soil to degradation [4]. Ecologically, soil aggregate stability reduces soil surface sealing and crusting, which are closely related to soil erodibility [5,6]. Therefore, its sympathy plays a crucial function in establishing environmental protection measures. Strongly aggregated soil is the result of the interaction of various factors such as soil organic carbon, soil texture, wet-dry cycles, etc. In this regard, SOC plays a key role in stabilizing soil aggregates [7]. Moreover, Imhoff et al. [8] reported that clay and silt fractions (<50 μm) behaved as cementing agents in water-stable aggregate formation. Furthermore, Amezketa [9] pointed out that aggregate breakdown by differential swelling increases with increasing clay content, which, however, contrasts with the results reported by Imhoff [8]. Generally, there are two main groups of factors influencing soil aggregate stability namely primary soil characteristics such as clay composition and organic matter and external factors such as dry-wet cycles [10]. However, few studies particularly investigated the impacts of external factors regardless of the effect of soil organic matter.

Soil aggregate stability can be quantified by applying forces likely similar to the ones they are exposed to in the field. The most common is the break-up of large aggregates into small aggregates classes by wet sieving method, which are expressed by Mean Weight Diameter (MWD), Geometric Mean Diameter (GMD) and Aggregate Stability percentage (AS%) indices. Many years ago, the wet sieving method was introduced by researchers and then improved for providing accurate results [11,12]. Considering its working conditions, this method is highly labor-intensive and time-consuming. Additionally, the wet sieving method has shown different limitations. Those include a lack of repeatability and a limited number of sieve sizes [13]. A Laser Diffraction (LD) has proved to be the most effective technique to determine soil particle size and aggregate stability of small aggregates (<2 mm). This has been introduced to overcome some of the inherent limitations related to most of the traditional methods. Erktan et al. [14] reported the existence of similar trends between the disintegration of soil aggregates (<1 mm) under stirring, and sonication within the LD and the disintegration of the aggregates (3–5 mm) after immersion in water. Due to its use of smaller aggregates and a few masses of the samples, an LD is the most advantageous method for evaluating soil aggregate stability [15]. Indeed, the LD method is deemed to be more appropriate, reproducible, precise, and accurate than previously-used methods for aggregate stability determination. The additional advantage of this method is that soil aggregates may be analyzed after the removal of soil organic matter and calcium carbonate. This indicates the effectiveness of LD method when determining the effect of soil organic matter and calcium carbonate on soil aggregate stability [16]. Using an LD, a number of previous studies presented the results of soil aggregate stability by Mean Volume Diameter (MVD) [17] and the cumulative distributions ($D_{10}$, $D_{50}$, and $D_{90}$) of aggregate sizes [14].

The spatial distribution of the above-mentioned indices possesses many outstanding interactive processes including natural ecological processes and intensive human activities, such as soil properties, land use type and landscape structure, topography, vegetation cover, hydrothermal conditions, and wet-dry cycles [18,19]. Throughout the interaction of matric water potential and soil particles, drying and wetting cycles can directly affect soil aggregate stability [20]. Due to shrinking and swelling caused by wet-dry cycles, the soils with a large amount of macro-aggregates develop more fissures along the plane of weakness [21], which consequently result in soil aggregates destruction. In addition, these cycles influence the soil properties, such as its strength and hydraulic stability, which results into cracking and stability failure [10]. The effects of drying and wetting cycles on soil aggregates may generally differ according to different factors including: (1) soil type, (2) length of cycles, (3) topography, (4) vegetation type, and (5) chemical properties of water.

The Three Gorges Dam implementation has created an artificial reservoir known as the Three Gorges Reservoir (TGR). Currently, the water-level of the TGR fluctuates from an altitude of 145 m

to 175 m. Purposely, the water-level is lowered in the wet season (May to September) for flood control and raised to its maximum level in the dry season (October to April) for hydroelectric power generation [22]. The water-level fluctuation significantly affects soil structure and composition in the riparian ecosystem [23]. In the WLFZ, the fluctuation of water influences the variations of different environmental processes and functions including soil ecosystem. The long period of inundation has remarkably depleted the vegetation cover in the TGR. Most native species were destroyed by long-lasting flooding in winter and summer drought, which consequently led to a considerable vegetation cover and community diversity decline. Only flood- and drought-resistant species become dominant in this area, with a dominance of annual herbs in the lower portion, perennials herbs in the middle portion, grass and shrubs in the upper portion, and trees in the higher portion [24]. The water-level fluctuation induces continuous wet-dry cycles (from 145–175 m), which may consequently influence the changes of soil aggregate stability and particle size distribution in the Water Level Fluctuation Zone (WLFZ). Having a great insight into aggregate stability changes would yield useful information on sustainable soil erosion mitigation, ecological restoration, and soil conservation measures in the riparian zone of the TGR. However, the long-run temporal variation of soil aggregate stability as influenced by external factors in WLFZ of the TGR has not yet investigated. Therefore, the objectives of this study were: (1) to investigate the particular effect of drying and wetting cycles on soil aggregate stability changes within time in the WLFZ, (2) to evaluate the temporal variation of grain size distribution, and (3) to identify the influence of soil organic matter on soil aggregate stability dynamic along the elevations in WLFZ.

## 2. Materials and Methods

### 2.1. Study Area

The Three Gorges Reservoir (TGR) is located between latitude 28°56′ N–31°44′ N and longitude 106°16′ E–111°28′ E covering the lower section of the upper reaches of the Yangtze River, with an area of $5.8 \times 10^4$ km$^2$ [25]. The current study sampled the riparian soils in Zhong County, Chongqing city, exactly from the middle section of the TGR on the right side moving from the Three Gorges Dam. In this region, there occurs a periodic water-level dynamic ranging from the elevation of 145 m to 175 m (Figure 1), and the area is referred to as the Water-Level Fluctuation Zone. Before the final impoundment, the water level raised to 135 m, 156 m, and 175 m in 2003, 2006, and 2009, respectively. The maximum water level in 2012 was still 175 m, which indicates that lower land experienced long period of inundation (from 2003 to sampling time) than upper land (from 2009 to sampling time) [26]. The study area is characterized by a subtropical monsoon climate, with a mean annual temperature of 18.2 °C and an annual precipitation of 1172.1 mm that is unequally distributed between seasons, with higher proportions of rain from May to September [27].

The soil of the study area is covered by sandstones, siltstones, and mud-stones of the Jurassic Shaximiao Group (J2 s) and is dominated by purple soil. The purple soil is a fast weathering soil type of the Jurassic rocks [28]. It is classified as an Orthic Entisol in the Chinese Soil Taxonomic System, a Regosol in the IUSS WG Taxonomy, and an Entisol in the USDA Taxonomy [29,30]. The topography of the area is dominated by both gentle and flat slope at each elevation, with a slope gradient of less than 5°, reflecting that soil erosion may unlikely influence various soil physicochemical behaviors (Table 1). The analyzed slope was created by an artificial embarkment long time ago, but it is rarely modified by sediment deposition due to the upper land soil erosion. The water-level fluctuation has significantly impacted the extinction of the pre-existing plant species including therophyte (*Digitaria ciliaris*, *Leptochloa chinensis*, and *Setaria viridis*), perennials (*Cynodon dactylon Capillipedium assimile*, and *Hemarthria altissima*), and ligneous plants (*Ficus tikoua*, *Pterocarya stenoptera*, and *Vitex negundo*), [31]. Presently, flood tolerant species (e.g., *Cynodon dactylon*) are largely distributed in the study area (Table 1).

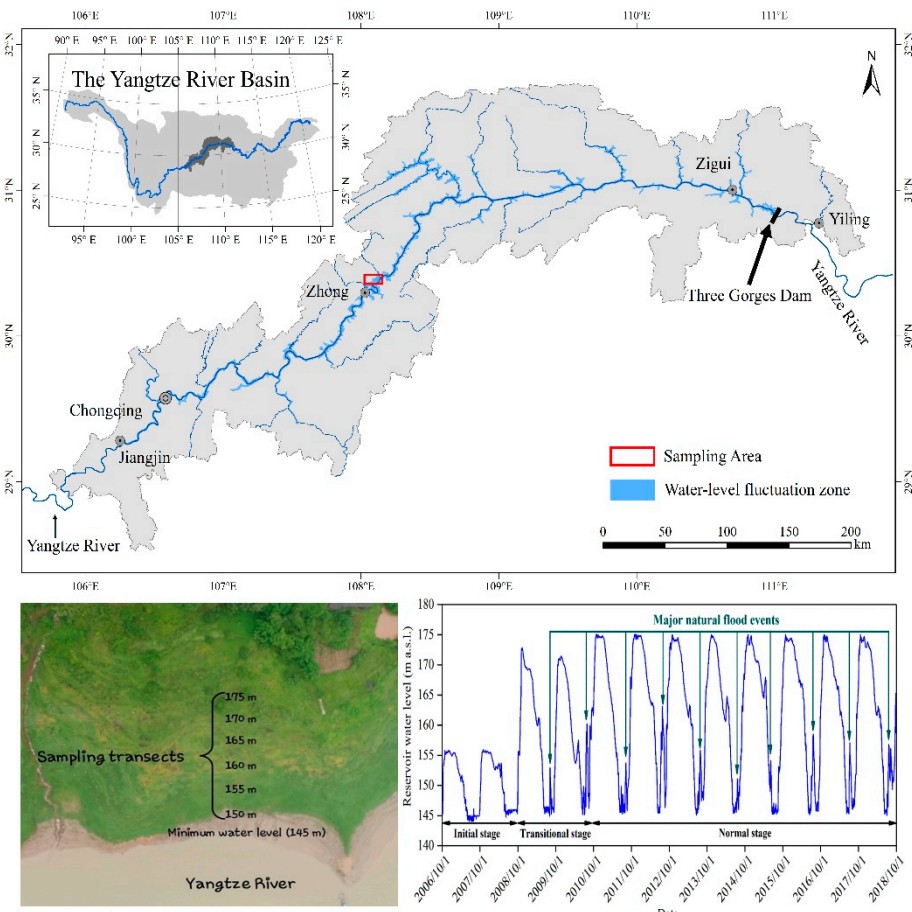

**Figure 1.** Description of the study area. (top) The upper Yangtze River basin and the enlarged map of the Three Gorges Reservoir indicating the sampling area; (lower left) aerial image of the study area; (lower right) seasonal water level-fluctuation in the Three Gorges Reservoir since 2006.

**Table 1.** General characteristics of sample plots along the selected elevations in the Water-Level Fluctuation Zone of the Three Gorges Reservoir.

| Elevation (m a.s.l) | Annual Inundation Time/Day | Inundation Depth (m) | Land Use | Soil Type | Slope Gradient/° | Main Vegetation Type |
|---|---|---|---|---|---|---|
| 150 | 288 | 25 | Grassland | Purple soil | 3 | Cynodon dactylon (L.) Pers., Alternanthera philoxeroides (Mart.) Griseb. |
| 155 | 250 | 20 | Grassland | Purple soil | 3 | Cynodon dactylon (L.) Pers., Alternanthera philoxeroides (Mart.) Griseb. |
| 160 | 212 | 15 | Grassland | Purple soil | 3 | Cynodon dactylon (L.) Pers., Xanthium sibiricum Patrin ex Widder |
| 165 | 152 | 10 | Grassland | Purple soil | 3 | Cynodon dactylon (L.) Pers., Xanthium sibiricum Patrin ex Widder, Digitaria sanguinalis (L.) Scop. |
| 170 | 100 | 5 | Grassland | Purple soil | 3 | Cynodon dactylon (L.) Pers., Xanthium sibiricum Patrin ex Widder, Digitaria sanguinalis (L.) Scop. |
| 175 | 3 | <0.5 | Grassland | Purple soil | 3 | Cynodon dactylon (L.) Pers., Xanthium sibiricum Patrin ex Widder, Digitaria sanguinalis (L.) Scop. |

*2.2. Sample Collection*

This study focused on the top soil of the riparian area (0–10 cm) in the Three Gorges Reservoir. Field sampling was conducted in early June in 2012 and 2016, when the water level was at its base level (145 m). Noting the 30 m vertical length of water-level fluctuation from 145 m to 175 m [31], the sampling plots were established at elevation of 150 m, 155 m, 160 m, 165 m, 170 m, and 175 m, respectively, taking into account an interval of 5 m. The annual inundation time, inundation depth, land use, soil type, slope gradient, and vegetation type at each elevation are presented in (Table 1). All samples were collected using stainless steel shovel after the removal of deposited surface sediments at elevations below 170 m. The layers of the removed sediments are described as 10 cm, 20 cm, 9 cm and 7 cm at 150 m, 155 m, 160, and 165 m, respectively. At each elevation, samples were collected from five points in a quincunx pattern within a sampling plot (5 × 5 m), then, the samples were thoroughly mixed to make a composite sample representing the study area; this was again split repeatedly by quartering until a sample weigh approximately 1 kg. Therefore, four different samples from the same elevation have been analyzed to determine the parameters measured in the present study. The samples were immediately packed into plastic bags and transported to the laboratory where they were air-dried at room temperature before manual removal of rock fragments and visible plant residues and then sieved to pass through a 2 mm sieve. Particular soil aggregate sizes (1–8 mm) were selected from 2016 soil samples for the wet sieving method [11].

*2.3. Soil Physicochemical Characteristics*

Soil organic carbon (SOC) was quantified by the wet oxidation method (Walkley and Black) using $K_2Cr_2O_7$–$H_2SO_4$ [32]. Soil organic matter (SOM) has been calculated using the Van Bemmelen factor. Consequently, the values of SOM were obtained directly by multiplying 1.724 to the values of SOC [33]. Soil particle size distribution (PSD) was determined using a Marvin Laser Particle size Analyzer (Masterizer 2000, Malvern Instruments Ltd., Worcestershire, UK) after treating samples with Hydrogen Peroxides, Hydrochloric acid, and Sodium hexametaphosphate for primary soil particles separation. The percentages of clay, silt, and sand were calculated basing on the volume percentage of the corresponding sizes according to the United States Department of Agriculture (USDA) PSD classification system.

*2.4. Aggregate Stability Tests*

2.4.1. Measuring Aggregate Stability by Laser Diffraction

The laser diffraction method was applied to analyze the stability of soil aggregates less than 2 mm by using a laser diffraction granulometer (Malvern Mastersizer 2000). Regular air-drying was applied for all samples to homogenize their moisture content. 0.5 g of soil aggregates were pre-treated with $H_2O_2$ to remove soil organic matter and then HCl for the removal of calcium carbonate [16]. Unlike for particle size distribution, sodium hexametaphosphate was not used during the analysis of soil aggregate stability with this method. Finally, the pre-treated samples were transferred to Marven Hydro 2000 unit (aqueous vessel), to allow them to circulate within the water in the measuring cell and then apply the ultrasonic dispersion for two minutes immediately prior to the analysis to certainly disrupt all aggregates. The readings were reported as percentages by volume. Immediate particle size distributions were repeatedly recorded, and the aggregate stability index was calculated using the well-known relationship of particle size ranges with their corresponding volume percentages.

The Mean Volume Diameter (MVD) index was calculated using Equation (1) [17,34]:

$$\mathbf{MVD} = \sum_{i=1}^{n} \overline{X}i \times V_i, \tag{1}$$

where $\overline{X}i$ is the mean diameter of each size fraction (μm) and $V_i$ is the volume proportion of the aggregates corresponding to that size fraction.



Soil aggregate breakdown was further evaluated by the changes in particle size distribution (PSD) along different elevations and within sampling years. The typical parameters used in this study are the PSD percentiles (10th, 50th, and 90th), namely $D_{10}$, $D_{50}$, and $D_{90}$, which means that 10%, 50%, and 90% of the sample contains grains with diameter less than or equal to D value.

### 2.4.2. Wet Sieving Method

Soil aggregate stability for sample collected in 2016 was determined by applying the technique similar to the method described by Yoder et al. [12]. Thirty grams (<8 mm) of air-dried soil sample was put on the set of sieves with opening sizes 5.0, 2.0, 1.0, 0.5, and 0.25 mm ordered from top to bottom. Sieves were then immersed into the water and allowed to rise and lower to 1.3 cm, 35 times/min, for 3 min. The remained aggregates on each sieve were collected, oven-dried, weighed, and finally used to calculate the water-stable aggregate indices. The mean weight diameter (MWD) and geometric weight diameter (GMD) were used to express the dynamic changes of soil aggregate stability in the Water-Level Fluctuation Zone. The above-mentioned indices were calculated by the Equations (2) and (3) [35]:

$$MWD = \sum_{i=1}^{n} \overline{Xi} \times Wi \,, \tag{2}$$

$$GMD = exp\left[\frac{\sum_{i=1}^{n} Wi \times \ln\overline{Xi}}{\sum_{i=1}^{n} Wi}\right], \tag{3}$$

where $\overline{Xi}$ is the mean diameter of size class i (mm) and $Wi$ is the mass percentage of aggregates in size class i.

It should be noted that the above equations represent the proportions of large macroaggregates and the size of the most frequent aggregate size from the samples [36].

### 2.5. Statistical Analysis

The t-test was used to examine the difference in grain size distribution, SOM, MVD, $D_{10}$, $D_{50}$, and $D_{90}$ between 2012 and 2016 soil samples. Pearson's Correlations were applied to determine the relationships among clay, silt, sand, and SOM for both 2012 and 2016 samples. A linear regression analysis was applied to examine the relationships between SOM with MVD, MWD, and GMD for the 2016 samples. On the other hand, the linear and polynomial regression models were used to identify the relationship between SOM, grain size at $D_{10}$, $D_{50}$, $D_{90}$, and elevations. Both *t*-test and Pearson's correlations were conducted with IBM SPSS 26.0 software package for Windows, while the regression analysis was carried out with RStudio, Version 1.2.1335 (2009–2019). In this study, a 95% confidence level ($p < 0.05$) was considered for all statistical analyses. All graphs presented in the present study were drawn using the SigmaPlot for Windows, version 14.0 (Systat Software, Inc, wbcubed GmbH, Germany).

## 3. Results

### 3.1. Soil Organic Matter Dynamic in the WLFZ

Soil organic matter (SOM) primarily act as a cementing agent for stabilizing soil aggregates.

By carefully examining the data, it was found that SOM content increased with an increase in the elevation, and that it was higher in 2012 than in 2016. Soil organic matter varied between 9.46 to 24.95 g $kg^{-1}$ in 2012 and 6.25 to 18.20 g $kg^{-1}$ in 2016, respectively. The statistical significance of the present results remained unchanged at $p < 0.05$ for the concentration of SOM irrespective of the mean soil organic matter concentration of purple soils in 2012 was approximately 1.5 times higher than in 2016 (Table 2).

**Table 2.** Pearson correlations between Soil organic matter (SOM), Clay, Silt, and Sand for both 2012 and 2016.

|  | Year | Clay | Silt | Sand |
|---|---|---|---|---|
| SOM | 2012 | 0.77 | 0.86 * | −0.89 * |
|  | 2016 | 0.52 | 0.91 * | −0.89 * |

* Significant correlation at $p < 0.05$.

A noticeable decrease of SOM from 2012 to 2016 was obviously recorded in lower and upper elevations, with a 8.54 gkg$^{-1}$ and 8.98 gkg$^{-1}$ decrease at 155 m and 170 m, respectively. As shown in (Figure 2), the trend of the results illustrates that SOM continuously increased along the elevation, with the exception at 165 m in 2012 and 170 m in 2016. This clearly indicates the negative effect of water logging on the distribution of soil organic matter.

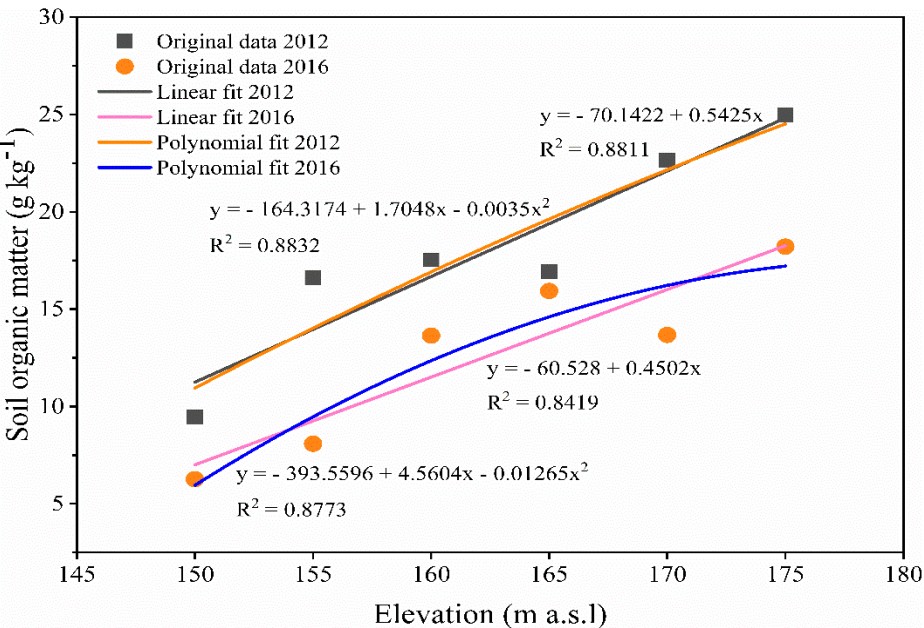

**Figure 2.** Variation of soil organic matter (SOM) by elevation in 2012 and 2016. The elevation is expressed as meter above sea level (m a.s.l). The prediction of SOM as the elevation increases is indicated by linear and polynomial fitting lines for both sampling years.

Soil organic matter peaked at 175 m for both 2012 and 2016 (Figure 2). Organic matter accumulation due to the decomposition of living organisms was greatly observed in higher elevations compared to lower elevations, arguing that short inundation duration and shallow inundation depth at higher elevations significantly contributed to the concentrations of SOM increment. Figure 2 further illustrates the two established regression models showing the relationships between SOM and elevations for 2012 and 2016 data. The results showed that polynomial fitting curves should provide the best predictive values of SOM along the elevations with $R^2 = 0.87$ compared to linear fitting line with $R^2 = 0.84$ for 2016 data. The upward direction of linear and polynomial fit lines explains the increase of SOM as elevation increases. Within the study period, there have been numerous hydrological alterations in the WLFZ, which possibly affected the SOM in multiple ways. Despite the other factors, plant deterioration should be the main reason declining soil organic matter from 2012 to 2016. In inundated area (poorly drained area), the accumulated dry living matter followed by low oxygen or anaerobic conditions inhibit organic matter decomposition [37]. Soil water interaction induces anaerobic chemical exchange, which consequently results in low organic matter decomposition. The upper land accounts higher vegetations than lower land because hydrological stresses destroyed extensively the vegetations in

the lower land. The rapid decomposition of the accumulated living organisms when water level is at 145 m followed by slow decomposition of organic matter when water level is at 175 m due to anaerobic situation contributed to the increase of soil organic matter in the upper land.

### 3.2. Soil Particle Size Distribution Characteristics in the WLFZ

Uneven distribution of particle size potentially influences various soil physical properties. Very fine grain size, such as colloids, plays a crucial role in stabilizing soil aggregates at a large scale. It is therefore important to understand the distribution of particle size and its impact on soil aggregate stability in the WLFZ. The prevalence of silt was highly observed in the study area with the fraction ranging from 69.52% to 75.58% in 2012 and 66.48% to 86.16% in 2016. Sand did not vary greatly, with the proportions ranging from 15.17% to 22.52% in 2012 and 10.84% to 32.05% in 2016, while clay proportions ranged from 7.47% to 9.85% in 2012 and 1.45% to 3.52% in 2016. Additionally, the surface soil of the WLFZ showed a relatively large statistical decrease in clay fraction ($p < 0.001$) and a significant increase in silt ($p < 0.05$). Despite the large changes in clay fractions, sand particles have slightly increased from 2012 to 2106 without any significant difference at $p < 0.05$ (Figure 3 and Table 3).

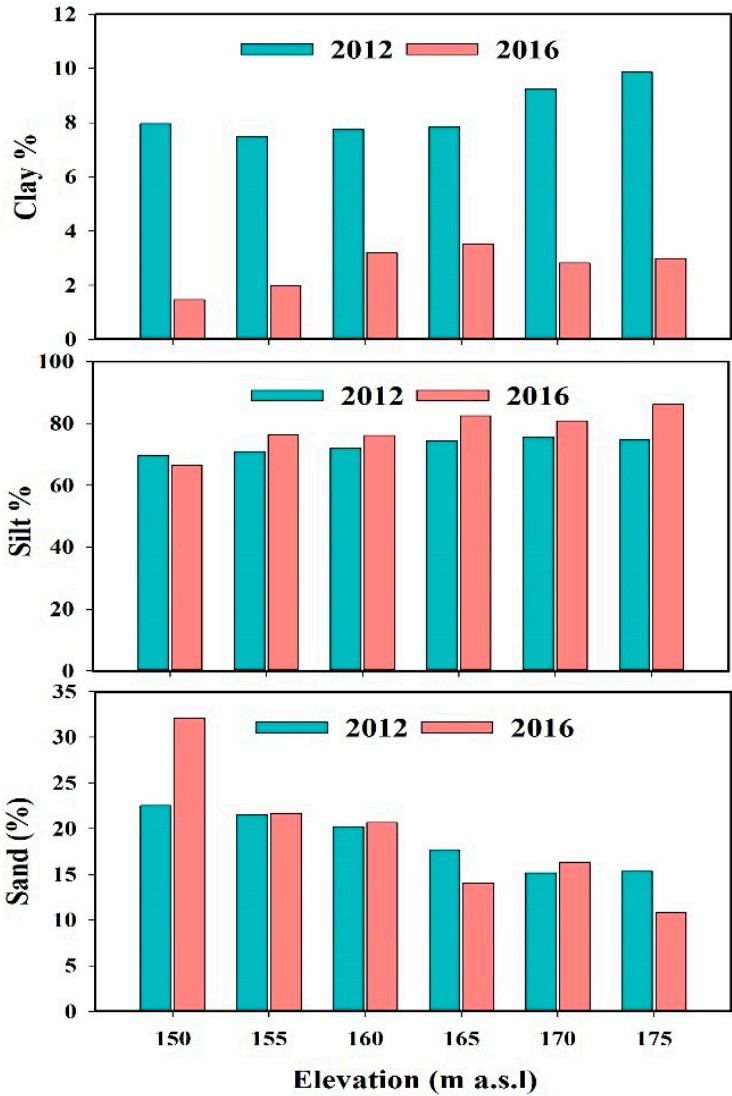

**Figure 3.** Soil grain size distribution with clay, silt, and sand percentages in the Water-Level Fluctuation Zone of the Three Gorge Reservoir (2012 and 2016).

**Table 3.** Summary statistics of independent t-test for MVD, $D_{10}$, $D_{50}$, $D_{90}$, Clay, Silt, and Sand between 2012 and 2016.

| Variable | Year | Means | t | df | p |
|---|---|---|---|---|---|
| MVD | 2012 | 3.50 ± 0.61 a | −2.46 | 10 | 0.03 |
|  | 2016 | 4.67 ± 0.98 b |  |  |  |
| $D_{10}$ | 2012 | 3.50 ± 0.61 a | −4.42 | 10 | 0.01 |
|  | 2016 | 4.67 ± 0.98 b |  |  |  |
| $D_{50}$ | 2012 | 17.49 ± 2.07 a | 0.54 | 10 | 0.59 |
|  | 2016 | 16.23 ± 5.22 a |  |  |  |
| $D_{90}$ | 2012 | 71.31 ± 10.83 a | −1.33 | 10 | 0.21 |
|  | 2016 | 87.11 ± 26.79 a |  |  |  |
| SOM | 2012 | 18.02 ± 5.40 a | 1.86 | 10 | 0.09 |
|  | 2016 | 12.63 ± 34.58 a |  |  |  |
| Clay | 2012 | 8.35 ± 0.95 a | 11.26 | 10 | 0.001 |
|  | 2016 | 2.65 ± 0.78 b |  |  |  |
| Silt | 2012 | 72.90 ± 2.40 a | −1.75 | 10 | 0.11 |
|  | 2016 | 78.08 ± 6.83 a |  |  |  |
| Sand | 2012 | 18.74 ± 3.14 a | −0.15 | 10 | 0.88 |
|  | 2016 | 19.25 ± 7.46 a |  |  |  |

Note: Means are presented with standard deviations. Values in the same column followed by similar lowercase letters are not significantly different.

In line with the findings of this study, clay proportions notably declined at 175 m, with 6.86% of decrease, while silt and sand proportions highly increased at 175 m and 150 m, with 11.35% and 9.53% of the total increase.

The distribution of clay fractions was highly recorded in upper elevations from 165 m to 175 m compared to lower elevations from 150 m to 160 m in 2012 and 2016. Figure 3 displays a little increasing trend of silt fractions along the elevation. Contrary, sand proportions revealed a decreasing trend ranked in the order of 150 m > 155 m > 160 m > 165 m > 175 m > 170 m in 2012 and 150 m > 160 m > 155 m > 170 m > 165 m > 175 m in 2016. In the course of this work, we observed an insignificant positive correlation of SOM with clay (r = 0.77) and a slight significant positive and negative correlation between SOM and silt (r = 0.86, $p < 0.05$) and sand (r = −0.89, $p < 0.05$) in 2012. On the other hand, a non-significant positive correlation between SOM and clay (r = 0.52) and a significant positive and negative correlations between SOM with silt (r = 0.91, $p < 0.05$) and sand (r = −0.89, p < 0.05) were found in 2016 (Table 2).

### 3.3. Temporal Variation of Soil Aggregate Stability in the WLFZ

The Three Gorges Reservoir is an artificial reservoir inducing a continuous special seasonal change due to water raising and lowering along the elevations in the water-level fluctuation zone. This can cause a relatively large shifts of wetting and drying in the riparian zone of the Three Gorges Reservoir within time. Currently, the lowest water level in the WLFZ is 145 m, and as the reservoir's water moves up, different storage periods occur at different elevations, and as the water goes down, drying arises at certain elevations. Therefore, it is crucially important to consider time and elevations when characterizing the variability of soil aggregate stability in the WLFZ.

Aggregate size distribution dynamic was characterized by three particle size distribution percentiles $D_{10}$, $D_{50}$, and $D_{90}$. Looking at the initial and final grain size in 2012 and 2016, respectively, there was no significant difference at $p < 0.05$ for $D_{50}$ and $D_{90}$. However, the PSD at $D_{90}$ showed a little increase from 2012 to 2016 compared to $D_{50}$. The PSD results at $D_{10}$ significantly increased ($p < 0.01$), indicating the coarseness of small aggregates in 2016 than 2012 (Table 3). Particle sizes at 10th, 50th, and 90th percentile ($D_{10}$, $D_{50}$, and $D_{90}$) variations for different elevations in WLFZ are displayed in (Figure 4). For 2016, particle sizes at $D_{10}$ and $D_{90}$ were coarser than the grain sizes observed in 2012, with the average size increase of 1.26 μm and 15.8 μm, respectively (Figure 4 and Table 3). The change

of grain size at all percentiles varied constantly for all elevations in 2012, contrasting with that found in 2016. For all PSD percentiles, the absolute variation of grain size was clearly observed for the lower elevations, reflecting the effect of inundation duration on soil aggregate size dynamic in the WLFZ. Similar phenomena induced a great increase of grain size at 150 m from 2012 to 2016, with a cumulative increase of 2.47 μm, 7.47 μm, and 28.81 μm at $D_{10}$, $D_{50}$, and $D_{90}$, respectively. In this study, the coarse grain sizes and likely small grain sizes were recorded from 150 m to 160 m (lower elevations) and 165 m to 175 m (upper elevations), respectively, suggesting that the stable soil aggregates were highly distributed at lower land compared to upper land.

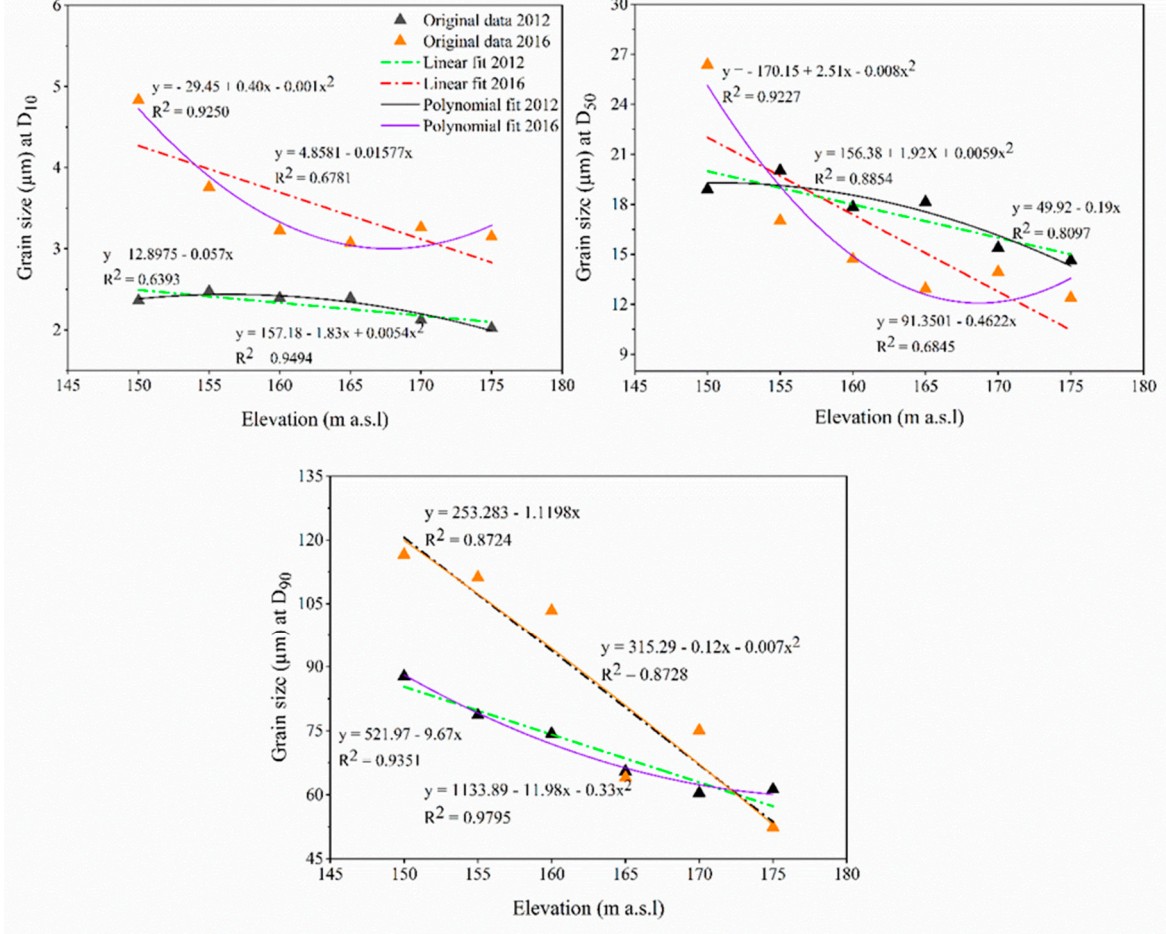

**Figure 4.** The status of soil particle size distribution at $D_{10}$, $D_{50}$, and $D_{90}$ indicating soil aggregate stability variation along the elevation ns and the elevational prediction of aggregate size at corresponding percentiles (10th, 50th, and 90th percentiles) by using linear and polynomial fitting models for 2012 and 2016.

The Mean Volume Diameter (MVD) ranged from 2.82 mm to 4.27 mm in 2012 and 3.32 mm to 5.88 mm in 2016, and the aggregate stability increases with the increase of MVD. Generally, the results of MVD showed a significant increase at $p < 0.05$ from 2012 to 2016, (Table 3). Due to higher MVD values observed in 2016, a strongly distributed soil aggregates were observed in 2016 compared to 2012, with an increase of MVD by 14.25% from 2012 to 2016. Figure 5 presents the decreasing trend of MVD in the WLFZ for 2012 and 2016 from the lower elevations to the upper elevations. At different elevations, soil aggregates responded differently to many wet-dry cycles that occurred within the study period. Although the overall status of soil aggregate in the WLFZ increased, the lower elevations have predominantly contributed to the increase of soil aggregate stability compared to

upper elevations, with the MVD increase of 23.11, 23.3, 15.78, 9.91, 20.47, and 7.06% at 150, 155, 160, 165, 170, and 175 m, respectively.

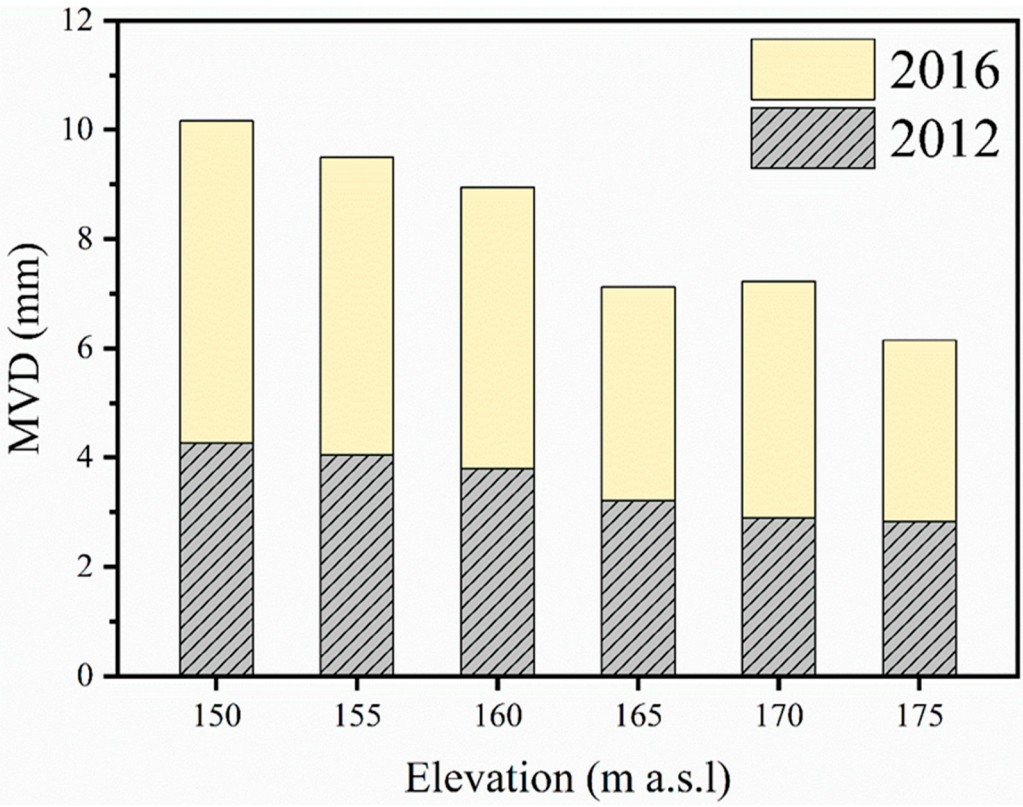

**Figure 5.** The Mean Volume Diameter (MVD) changes along the elevations in the Water-Level Fluctuation Zone of the Three Gorges Reservoir.

## 4. Discussion

### 4.1. Long-Time Effect of Wetting and Drying Cycles on Soil Aggregate Stability in WLFZ of the TGR

The external and internal forces resulting from the interaction of soil and water influence the mechanism of aggregation [1]. In the WLFZ, the artificial and natural wet-dry cycles induce a periodic soil-water interaction that should impact soil aggregate stability in TGR. Therefore, the present study hypothesized a significant difference in soil aggregate stability within five years (from 2012 to 2016). After rigorous examination, the results of the independent *t*-test showed a significant increase of Mean Volume Diameter ($p < 0.05$) between 2012 and 2016, reflecting an increase in soil aggregate stability between initial and final sampling years.

Mulia [38] reported a significant seasonal increase and decrease of aggregate stability, primarily attributed to precipitation and winter freezing-thawing cycles. In the light of reported findings, it is conceivable that seasonal dynamic change (increase and decrease) of soil aggregate stability occurring within a year influences the overall status of soil aggregate stability in the WLFZ of the Three Gorges Reservoir.

As shown in (Figure 5), the MVD in 2016 is higher than that in 2012, with an average increase of 3.50 mm in 2012 and 4.67 mm in 2016 (Table 3). This indicates that soil aggregates partially resisted the hydrological stresses and became more resistant to slaking as the wetting and drying cycles increases within the study period (from 2012 to 2016). The above results are in substantial agreement with those found by Denef et al. [39], stating that aggregates became stable and slake-resistant after two dry-wet cycles. Moreover, similar results were also achieved by Bravo-Garza et al. [40], where unamended soil (Vertisols) under wetting and drying cycles showed an increase of 20% in

the amount of small water-stable macroaggregates (0.025–2 mm) compared to amended samples after one wetting and drying cycle. However, the study of Allison [41] and Tisdall et al. [42], in contrast to our results, showed that wetting and drying cycles decreased macro-aggregate stability. Repeated cycles of drying and wetting play a major role in aggregation through shrinkage and swelling that lead to the formation of soil aggregates [1]. It appears, therefore, that wetting and drying cycles affect aggregates through various processes, namely swelling and shrinkage of clays, physical transport, deposition, and hardening of organic and inorganic binding agents as well capillary stresses [43]. However, the present study excluded the effect of organic and inorganic binding materials. It has particularly considered the physical impact caused by wetting and drying cycles on soil aggregate stability in the WLFZ of the TGR. Those cycles orient fine particles at the extent they stay close each other so that the forces between them hold the particles firmly when they experience drying [44]. In addition, the alternating shrinkage and swelling due to wet-dry cycles enhanced the formation of soil aggregates resisting to further stresses. Generally, the findings of this study reveal that the MVD has significantly increased from 42.86% in 2012 to 57.12% in 2016. In practice, structural stability assessed within different time frame explains the combination of various processes, such as cementation resulted from the presence of SOM, microbial activities influenced by soil water content variation, and crystallization of solutes when the concentration of the solutes in the pore fluid closely approaches the solubility limit prior to the onset of drying [23]. As the current study did not consider the effect of soil organic matter, wetting and drying cycles have predominantly increased aggregate stability in the water level fluctuation zone of the TGR by forming aggregates which are more resistant to slaking.

Compared with the traditional methods, characterizing soil aggregate stability using an LD provides not only accurate and reproducible results but also offers various alternatives for presenting the results of soil aggregate stability. In essence, this method involves various disaggregation processes, namely (1) slaking, (2) collision of aggregates during their movements inside the Hydro 2000 unit, and (3) sonication during the determination of soil aggregate stability. Apart from MVD, aggregate stability has been identified by using the percentiles $D_{10}$, $D_{50}$, and $D_{90}$. Similar to MVD, most of the percentiles showed higher grain sizes in 2016. A recent study conducted by Erktan et al. [14] pointed out that the soil samples characterized by high aggregate stability presented the higher values of $D_{10}$, $D_{50}$, and $D_{90}$, while less stable aggregates showed the lower values for those percentiles. The TGR exhibits a continuous periodical wetting and drying cycles with respect to seasons (winter and summer). The present study investigated the temporal variation of soil aggregate stability being affected by wetting and drying cycles occurring in the WLFZ of the TGR. However, it did not consider the changes in soil aggregates by counting the number of wetting and drying cycles.

Within this study, the variation of aggregate size at different percentiles indicated a significant increase at $D_{10}$, a slight increase at $D_{90}$, and a decrease at $D_{50}$ from 2012 to 2016 as displayed in (Figure 4). The above results suggest an increase in soil aggregate stability, with the 2016 soil samples being characterized by the higher values of $D_{10}$ and $D_{90}$ compared to 2012 samples. As the wetting and drying cycles increases, the sizes became coarser for D10 and $D_{90}$, with an increase of 21.49% and 9.97%, respectively. Our findings highlight that the more the wet-dry cycles increase in the WLFZ, the higher the soil aggregate stability becomes more resistant to slaking. This is in line with the recent studies. Thus far, Xu et al. [10] noticed that most aggregates were more resistant to slaking after two cycles. Microstructures rearrangement of soil aggregates is potentially enhanced by the increase in the number of dry-wet cycles, which eventually result in a new equilibrium state of microstructures [45]. In general, several studies reported that wet-dry cycles significantly influence water stable aggregates. Despite that some highlight an increase [10,45] and a decrease [20,46] of soil aggregate stability according to the historical background and the nature of soil for the study area.

Results show that clay proportions decreased significantly from 2012 to 2016 (Figure 3) and this indicates that the lower concentrations of clay fractions in 2016 have significantly increased aggregate stability. The possible reason for this should be the aggregate breakdown caused by internal pressure through clay differential swelling. Although Wei et al. [17] indicated that clay plays a crucial role

in stabilizing aggregates, Amezketa [9] noticed that aggregate breakdown by differential swelling increases with the increasing of clay content. In this regard, the observed increase in soil aggregate stability might have been attributed to lower clay proportions. For this changed ecosystem, the present findings provide and explain basically how continuous field wet-dry cycles sustained soil aggregate stability in the riparian zone of the TGR which may consequently be attributed to crusting reduction, increase of the infiltration, and finally environmental and ecological protection from degradation [3]. Weather conditions, mainly rainfall and temperature, are considered to be the factors affecting the variation of soil aggregate stability [47]. To assign this to the present results, the rainfall exerted pressure to soil aggregates was minimized by the vegetation cover, which consequently reduces its effect to aggregate disintegration compared to long lasting inundation period. On the other hand, temperature plays a crucial role in drying the soil after a long period of wetting.

### 4.2. Effect of Hydrological Regime on Soil Aggregate Stability Changes in the WLFZ

Historically, the Three Gorges Reservoir impounded periodically, with 135 m, 156 m, 172 m, and 175 m impounded in 2003, 2006, 2008, and 2010, respectively [48]. This clearly shows how soils at lower elevations stayed for a long time under hydrological stresses compared to upper elevations. In the WLFZ, the hydrological stresses vary within the elevations due to different inundation duration at each elevation. The inundation duration generally decreases with increasing elevation (Figure 6).

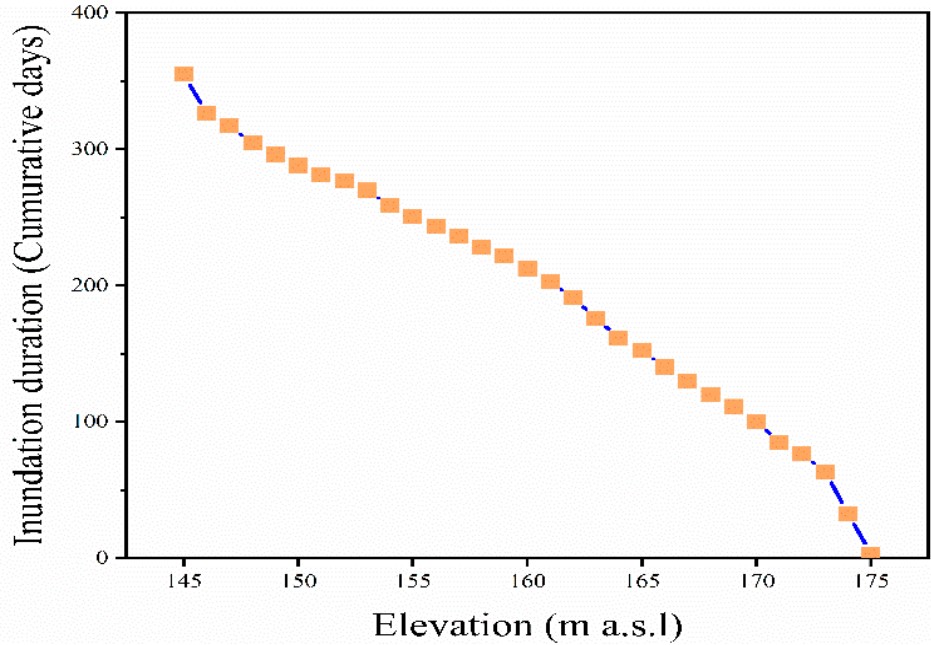

**Figure 6.** The vertical dynamic in inundation duration at each elevation in the WLFZ of the Three Gorges Reservoir.

The MVD, $D_{10}$, $D_{50}$, and $D_{90}$ slightly declined as the elevation increases, with higher values at lower elevations and lower values at higher elevations (Figures 4 and 5). The observed relationship between the elevations and hydrological changes of inundation duration highlights the impacts of different flooding intensities on soil aggregate stability in the riparian zone. Higher aggregate stability in the lower land of the riparian zone is related to the longer inundation duration. This may be partially explained by the effect of water content on soil aggregate stability. There are multiple potential explanations for this: one is the relationship between water content and stability resulting from age-hardening, which affects particles rearrangement and reinforces the links through redistribution of solutes and cementation [49]. The above relationship is likely to be complex due to the fact that various processes interact over different time scales [50]. In addition, the fact that the higher area

was in the very initial period of soil transformation caused by wetting may significantly explain its lower aggregate stability compared to lower areas experiencing several wettings and drying cycles. Contrarily, Cui et al. [51] and Jiang et al. [52] indicated lower Mean Weight Diameter (MWD) in the top two layers (0–10 and 10–20 cm) of the lower land. This may suggest a decrease in soil aggregation from the upper to lower elevations in the WLFZ. Stepping back to our results, the fact that the upper elevations experience more human activities such as tillage and conservative measures should explain the decline in soil aggregate stability at the upper elevations. Removal of SOM from the samples while determining aggregate stability by an LD may further explain the decrease of aggregate stability along the elevations. This has completely excluded the effect of SOM on soil aggregate stability. Instead, the above-mentioned studies considered the impacts of SOM on soil aggregation. The percent change in MVD, $D_{10}$, and $D_{90}$ between 2012 and 2016 was significant at lower land areas, and increased by 23.11, 23.3, 15.78, 9.91, 20.47, and 7.06% for MVD, 32.77, 18.12, 9.99, 9.04, 15.11, and 14.95% for $D_{10}$, and 30.39, 38.91, 26.04, $-1.50$, 15.47, and $-9.32\%$ for $D_{90}$ at altitudes of 150, 155, 160, 165, 170, and 175 m, respectively. The possible prediction of aggregate sizes along the elevations at all particle percentiles ($D_{10}$, $D_{50}$, and $D_{90}$) is presented in (Figure 4). Compared to linear fitting, the polynomial fitting curves have shown a higher fitting ability of grain size at all elevations between 150 m and 175 m. However, at $D_{90}$, the linear fitting lines almost coincide with the polynomial fitting curves. This indicates the predicting power of a linear relationship of grain sizes and elevations though the polynomials present a slight difference in coefficient of determination, $R^2 = 0.93$ in 2012, 0.87 in 2016 and 0.97 in 2012, 0.87 in 2016 for linear and polynomial fitting, respectively. Both linear and polynomial relationships presented a dawn ward curves, which arguably show a decline of grain size along the elevations (Figure 4). Sediment deposition in the lower land (below 165 m) has partially precluded a direct effect of the hydrological stresses on soil aggregates. In this regard, the aggregates in the upper land have directly experienced the natural and artificial hydrological disturbances, which potentially lead to disaggregation. Further factors, such as plant roots and weather conditions, contribute significantly to the amelioration of soil structures [47,53]. Although this study did not consider the effect of several grass types, densely vegetation distribution at the upper elevations as the result of late impoundment may affect soil structures. Roots exudates stabilize soil aggregates through the interaction of physical, chemical, and biological processes [53]. Higher vegetation cover at the upper elevations may explain higher values of MWD and GMD as the indices used to express soil aggregate stability.

Further results of aggregate stability by using a wet sieving method provided results indicating the increase of Mean Weight Diameter (MWD) and Geometric Mean Diameter (GMD) from lower to upper elevation in the WLFZ (Table 4). Different to LD results, wet sieving method has recorded the influence of soil organic matter. Figure 2 shows a decrease in soil organic matter, while Figure 5 presents an increase of MVD from 2012 to 2016. With this discrepancy, it may be attributed that SOM did not play any significant long-term role in stabilizing soil aggregates in WLFZ. This may or may not be true, but there is little logic based on disregarding the influence of soil organic matter during the determination of aggregate stability by Laser Diffraction in the present study. Moreover, Figure 7 presents the relationships between SOM, MVD, MWD, and GMD calculated for samples collected in 2016. The increase of MWD and GMD was mainly attributed to the increase of SOM with $r^2 = 0.89$ ($p < 0.01$) and $r^2 = 0.90$ ($p < 0.01$), while the increase of MVD was highly predicted by the decrease of SOM with $r^2 = 0.88$ ($p < 0.01$). The removal of SOM while analyzing aggregate stability may further explain the contradiction between the resulted MVD, and the MWD and GMD. Regardless of the effect of SOM, this study shows that strongly stable aggregates were most highly distributed in the lower elevations compared to upper elevations (Figure 5). Generally, aggregate stability increase ensures the durability of soil to several forces and other problems detrimental to soil environmental conditions. Undermining the effect of chemical bindings, the long-term dry-wet cycles occurred in lower elevations played a key role in sustaining soil cohesive forces which in turn maintain the soil resistivity to different environmental disturbances, thereby holding their shapes, sizes, pore structures, and soil organism relationship.

**Table 4.** Soil water-stable aggregate indices, MWD (Mean Weight Diameter), and GMD (Geometric Mean Diameter) in water-Level Fluctuation Zone for 2016.

| Elevation (m) | MWD (mm) | GMD (mm) |
|:---:|:---:|:---:|
| 150 | 1.42 ± 0.28 a | 0.61 ± 0.19 a |
| 155 | 2.46 ± 0.28 b | 1.32 ± 0.29 b |
| 160 | 3.48 ± 0.22 c | 2.11 ± 0.25 c |
| 165 | 3.81 ± 0.18 d | 2.53 ± 0.25 d |
| 170 | 3.91 ± 0.14 d,e | 2.83 ± 0.14 e |
| 175 | 4.09 ± 0.24 e,f | 3.08 ± 0.39 e,f |

Note: The results are presented as mean ± standard deviation. Values in the same column followed by different letters indicate significant difference at $p < 0.05$.

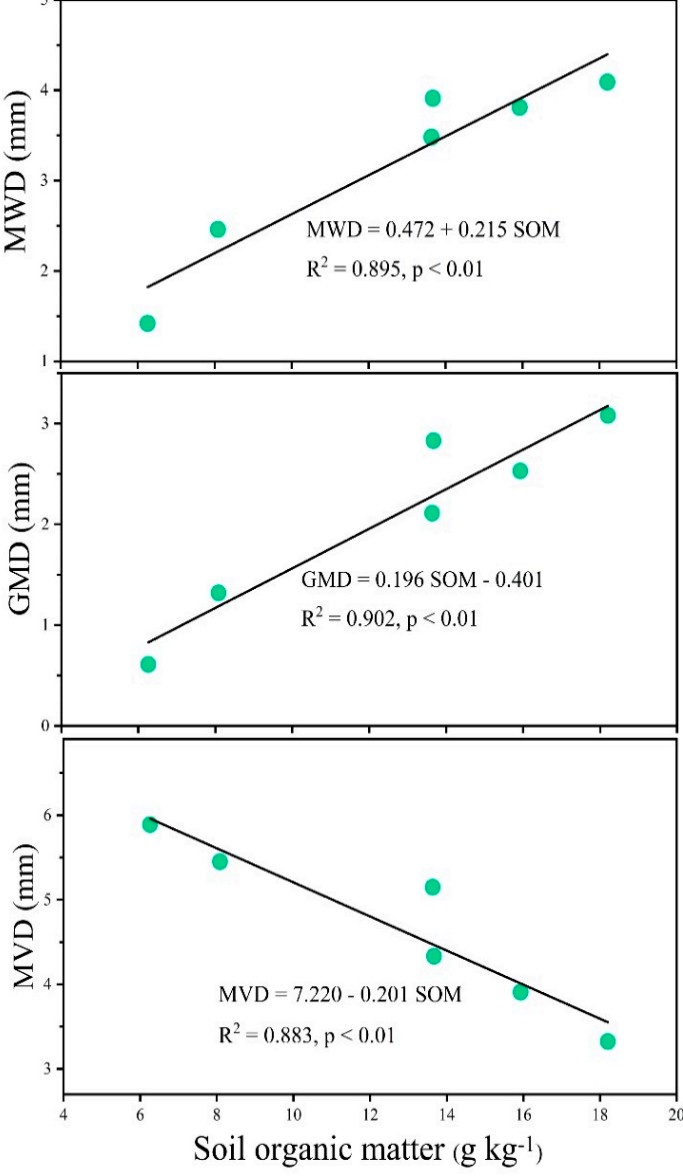

**Figure 7.** Relationship between soil organic matter and aggregate stability indices (**upper**) Mean Weight Diameter (MWD), (**middle**) Geometric Mean Diameter (GMD)), and (**lower**) Mean Volume Diameter (MVD) in the WLF Zone of the Three Gorge Reservoir (2016).

*4.3. The Hydrological Regime and Soil Organic Matter Impact on Grain Size Distribution in the WLFZ of the Three Gorge Reservoir*

Soil particle size distribution plays a key role in the classification and estimation of many soil properties [54]. Predominately, clay content, and types impact soil pore size distribution [55]. Previous studies on PSD in the WLFZ of the TGR concentrated on sediment PSD [22,56]. In order to address the previously unaccounted work, our study has basically investigated the temporal variations of soil aggregate stability and particle size distribution in WLFZ. Results evidently show a significant decrease ($p < 0.001$) of clay particles from 2012 to 2016 (Table 3), contrasting with a significant increase observed in silt particles ($p < 0.05$). Repeated wetting and drying cycles have consequently resulted in fine particle displacement in the WLFZ. Over a long period of time, pedological processes such as mineral weathering and illuviation are proposed to alter soil texture of a given area, and this may also be encouraged by mixing the pre-existing soil texture with the soil materials of different soil textural class [37].

Wetting followed by drying may certainly expose suspended sediment particle size on the surface soil. Thus, in addition to parent material weathering, alternating wetting/drying increased the deposition of silt particles, meanwhile decreasing fine particles within the WLFZ of the TGR. Silt fractions increased with the increase of the elevations and this suggests a negative relationship between silt distribution and inundation period. In 2016, clay and silt particles were positively correlated with SOM (r = 0.52 for clay and r = 0.91 (at $p < 0.05$) for silt), while sand was negatively correlated with SOM (r = − 0.89 (at $p < 0.05$)) (Table 2). This shows that the proportions of silt particles increased with the increasing of SOM, whereas sand proportions decreased with the increase of SOM in the study area. Six et al. [57] revealed a strong positive correlation between soil organic matter and clay particles. In normal conditions, there is a strong relationship between clay proportions with soil organic carbon. However, this condition is not common globally. Fine particles protect soil organic carbon from chemical weathering [58]. A strong positive correlation of silt with soil organic matter in our results is because silt particles are largely distributed in the study area. Therefore, they play a more significant role than clay as well as sand, which recorded a negative correlation with soil organic matter.

## 5. Conclusions

The regulation of water by the Three Gorges Dam (TGD) has induced a continuous water-level fluctuation in the TGR. This has consequently resulted in the variability of the riparian soil's physical characteristics within time and elevations. This study evaluated the long-run temporal dynamic changes of soil aggregate stability and particle size distribution in the riparian zone of the TGR using an LD. The study has particularly identified the physical changes of soil aggregate caused by wet-dry cycles regularly occurring in the WLFZ of the TGR, thereby disregarding the influence of chemical binding materials in soil aggregate formation. However, the comparative results of LD method and wet sieving method showed a trending difference of soil aggregate stability along the elevation because wet sieving has considered the effect of soil organic matter on aggregate stability. With the increase of the MVD from 42. 86% in 2012 to 57.12% in 2016, this study revealed that the aggregate stability in the WLFZ of the TGR increased, suggesting that continuous wetting and drying cycles significantly enhanced the stability of soil aggregates within the study period. Moreover, PSD percentiles, $D_{10}$, and $D_{90}$ presented the increase in aggregation as their sizes became coarser within time.

Over five years (2012–2016), a significant decrease ($p < 0.001$) and increase ($p < 0.05$) was found in very fine grain size (clay) and silt particles, respectively. On the other hand, the observed changes in coarse particles (sand) were not statistically significant. This explains the effect of drying on soil PSD variation, mainly attributed to the deposition of suspended sediment particle sizes. Based on the findings of this study, it can be concluded that natural wetting and drying cycles influenced the variability of soil aggregate stability and PSD in the WLFZ. This study basically investigated the long-term change in aggregate stability under the influence of external factors. We suggest that future studies should examine the combination of both internal factors or soil primary characteristics

and external factors to evaluate seasonal and yearly aggregate variability. Overall, the present study provides useful information for sustainable utilization of soil resources sustainable land management in ecologically vulnerable areas.

**Author Contributions:** Conceptualization, G.N. and Y.B.; methodology, G.N., Y.B., and X.H.; supervision, Y.B. and X.H.; software, G.N.; formal analysis, G.N., Y.B., and X.H.; data curation, G.N.; writing—original draft preparation, G.N.; writing—review and editing, G.N., Y.B., X.H., J.d.D.N., M.W., L.Y., J.L., S.Z., and D.K.; funding acquisition, Y.B. All authors have read and agreed to the published version of the manuscript.

**Funding:** Financial support for this study was jointly provided by the National Natural Science Foundation of China (Grant No. 41977075, 41571278 and 41771321) and the Sichuan Science and Technology Program (2018SZ0132).

**Acknowledgments:** The authors would like to thank all persons involved in field investigation and data preparation of this study.

**Conflicts of Interest:** The authors declare no conflict of interest.

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
