# Peer review of "Impacts of Water Level Fluctuations on Soil Aggregate Stability in the Three Gorges Reservoir, China"

_sustainability, doi:10.3390/su12219107_

Round 1

Reviewer 1 Report

I'm fully satisfied with corrects have been made in the paper. In my opinion, the analysis of soil agregates is one of the most difficult laboratory activities. It is very easy to change their original shape, size and layout during the soil prepration processes and analysis itself. Thanks to the authors for making appropriate comments in this regard. I am also pleased with the information on important dates for the functionality of the described area.

Author Response

Response to Reviewer #1’s Comments and suggestions

General comment: I'm fully satisfied with corrects have been made in the paper. In my opinion, the analysis of soil agregates is one of the most difficult laboratory activities. It is very easy to change their original shape, size and layout during the soil prepration processes and analysis itself. Thanks to the authors for making appropriate comments in this regard. I am also pleased with the information on important dates for the functionality of the described area.

Response: We highly appreciate the reviewer’s point of view after further revisions of the paper according to his/her comments and suggestions. The reviewer’s current comments are scientifically very encouraging, informative, and motivational. In fact, the authors conducted research in one of the most changing environments in China (Three Gorges Reservoir). It is quite interesting for scientists to continuously understand the variability of soil physical properties of this area and publish their results. We hope that the results of the present manuscript will be essential for future related works and environmental sustainability-related decisions in the study area and any other area with a similar situation as the Three Gorges Reservoir.

Reviewer 2 Report

Although the authors have made some changes to the manuscript, I recommend to reject it again due to the following reasons:

- Aggregate stability is highly variable. Therefore, two measurement campaigns (one in 2012 and one in 2016) with only one composite sample (taken in a plot of 5x5m) for each elevation step are not enough. For a correct analysis, at least 3-5 positions around the reservoir at each campaign should have been sampled. The authors claim that “4 different samples from the same elevation have been analyzed to determine the parameters measured in the present study” (l 158-160). But these four samples are only subsamples of the one sampled plot. The database for the following analyses is therefore far too small to obtain a scientifically sound result.

- Again, sample preparation procedure for aggregate stability measurement described in 2.4.1 seems to be wrong. To measure the soil aggregate stability, it is not meaningful to remove the soil organic matter and the calcium carbonate before the measurements, as they are important factors for soil aggregate stability. The authors replied: “Aggregate stability measurement described in 2.4.1 has also helped to basically understand the particular impacts of external factors such as dry-wet cycles instead of considering the internal factors such as soil organic carbon on soil aggregate stability. Therefore, the increase of soil aggregate stability within time in the study area is mainly linked to the continuous dry and wet cycles occurred in this area according to our results.” I cannot follow this argumentation and separation into external and internal. Organic carbon is very important for dry-wet-cycles and is strongly associated with aggregate stability. If you remove the organic carbon, you change the size and stability of the (natural grown) aggregation.

- The third point from the last review-process “In addition, further effects on soil aggregation (e.g. root growth, weather conditions) must be considered in the analysis.” was not considered in the manuscript. I wonder why not, for instance in the discussion section.  

Author Response

Response to Reviewer #2’s comments and suggestions

Comment 1: Aggregate stability is highly variable. Therefore, two measurement campaigns (one in 2012 and one in 2016) with only one composite sample (taken in a plot of 5x5m) for each elevation step are not enough. For a correct analysis, at least 3-5 positions around the reservoir at each campaign should have been sampled. The authors claim that “4 different samples from the same elevation have been analyzed to determine the parameters measured in the present study” (l 158-160). But these four samples are only subsamples of the one sampled plot. The database for the following analyses is therefore far too small to obtain a scientifically sound result.

Response: Thank you for this comment. The main objective of this research was to evaluate the long-term impacts of water level fluctuation on soil aggregate stability in the Three Gorges Reservoir. Therefore, five years difference was considered to significantly provide the overall image of soil aggregate stability variability in the study area. As presented in the Discussion section (pages 11 and 12/ lines 350-354), a seasonal and yearly decrease and increase of soil aggregate stability have been arisen, which may eventually help to overall conclude the 14.25% increase of MVD within five years. On other side, greater than 70% of the soil in the study area is purple soil. This explains a higher similarity of soil properties in the Three Gorges Reservoir (TGR). The variability of the TGR’s soil is basically associated with plant distribution. Particularly, our study accounted for sampling from the same grass type, which means that non sparsely (<3 m) sampling points may not be wrong to present the results of this study. The authors humbly recognized the reviewer’s comments and suggestions. We therefore recommended future studies to consider seasonal and yearly based data and sparsely sampling position along elevation to deeply cover highly variability of soil aggregate stability (page 18/lines 556-560). More interestingly, those future results will be matched with the results of this paper to provide the real status dynamics of soil aggregate stability in the TGR.

Comment 2: Again, sample preparation procedure for aggregate stability measurement described in 2.4.1 seems to be wrong. To measure the soil aggregate stability, it is not meaningful to remove the soil organic matter and the calcium carbonate before the measurements, as they are important factors for soil aggregate stability. The authors replied: “Aggregate stability measurement described in 2.4.1 has also helped to basically understand the particular impacts of external factors such as dry-wet cycles instead of considering the internal factors such as soil organic carbon on soil aggregate stability. Therefore, the increase of soil aggregate stability within time in the study area is mainly linked to the continuous dry and wet cycles occurred in this area according to our results.” I cannot follow this argumentation and separation into external and internal. Organic carbon is very important for dry-wet-cycles and is strongly associated with aggregate stability. If you remove the organic carbon, you change the size and stability of the (natural grown) aggregation.

Response: Thank you! The authors are pleased to provide more clarification for this comment. Indeed, one of the hypotheses for our study is that the water level fluctuations cause a decline in soil organic matter due to plant deterioration. Due to continuous wet-dry cycles, the direct effect of organic matter reduction within time is scientifically understandable but we also need to investigate the effect of other factors continuously affecting soil aggregate stability in the study area. Therefore, we needed to profoundly understand what should happen on soil aggregate stability when considering external factors (dry-wet cycles resulted from water level fluctuation) rather than basing on the total effect of soil organic matter as it is expected to decrease within a long period of time. That's why we removed soil organic matter from the samples before testing soil aggregate stability in order to identify the particular impacts of wet-dry cycles in the Water Level Fluctuation Zone of the Three Gorges Reservoir Area. As presented in the text (page 2/lines 59-61), internal factors influencing soil aggregate stability are described as soil inherent properties such as soil organic matter, cation exchange capacity, clay mineralogy, etc. On the other hand, external factors are directly related to climatic conditions and human activities such as wet-dry cycles, freeze-thaw cycles, irrigation, tillage, etc. In this study, we found that numerous wet-dry cycles rearrange and form aggregates resistant to slaking and hydrological stresses (page 12/lines 374-377). Generally, the additional information explaining how wetting and drying cycles may increase aggregate stability is found on (page 13/ lines 382-385).

Comment 3: The third point from the last review-process “In addition, further effects on soil aggregation (e.g. root growth, weather conditions) must be considered in the analysis.” was not considered in the manuscript. I wonder why not, for instance in the discussion section. 

Response: We kindly appreciate this helpful reviewer’s suggestion. Actually, root growth described as plant roots and weather conditions affect the variability of soil aggregate stability. The authors addressed this important suggestion in the Discussion section, thereby inserting some additional references and the author’s own statements. That information is tracked in the text (page 14/ lines 469-475 and page 13/ lines 422-426). However, we did not strongly insist on root growth effects because the present study did not sample from different grass types to identify their impacts on soil aggregate stability. This should be an interesting future study. Thank you very much.

Reviewer 3 Report

It's possible de compare your results with sites around the world just to see if the results could be genralized to other sites studies

Author Response

Response to Reviewer #3’s Comments and suggestions

Comment 1: It's possible de compare your results with sites around the world just to see if the results could be genralized to other sites studies.

Response: Thank you very much for assigning this important comment. The authors are grateful to address this awesome comment. In the text, some references providing the global results similar to the present study were added (page 13/ lines 409-411). However, some present a decline and an increase of soil aggregate stability under wet-dry cycles due to the historical background and soil composition of the selected study area. Therefore, generalization should not be efficient at all.

Round 2

Reviewer 2 Report

The data base is to small to dervie any scientifically sound result due to the fact that soil aggregate stability is highly variable. For future sampling design, I recommend to sample at least 3-5 positions for each investigated category.

This manuscript is a resubmission of an earlier submission. The following is a list of the peer review reports and author responses from that submission.

Round 1

Reviewer 1 Report

I find the paper very interesting. Changes in soil of riparian areas as a result of flooding the reservoirs with water are rarely reported. Please read my comments in the text carefully. Especially it should be added some information in the Methods and in the following text there should be precisely separated the indications concerning the grain and aggregates composition.

In my opinion, too little attention has been paid to the problem of different time of impact of the described factors on soils located in different positions of the transect. The reservoir was filled to the next ordinate for a number of years, and in addition, later also the fluctuations in the filling level were noted. In my opinion, this is the main factor causing the described results.

Reviewer 2 Report

The Authors present a study aiming to investigate the long-run temporal variation of soil aggregate stability as induced by water-level fluctuations in the riparian zone of the Three Gorges Reservoir, China.

The subject is certainly of interest and within the scope of the Journal. In addition, the manuscript is well-organized, the objective of the manuscript is clear and concise, the title fully reflects the contents. Moreover, the manuscript presents the results of the analysis in a logical manner, the figures are clear, easy to follow and informative.

In the reviewer's opinion, the present work provides useful information on the subject of which it treats. However, given the specificity of the work, the manuscript should be considered as a Case Study.

Reviewer 3 Report

The topic of this manuscript may within the scope of sustainability. However, I recommend to choose a more water or soil science related journal. 

The idea of the presented manuscript is interesting. However, as the presented data base is extremely low (described in l 119-129). Aggregate stability is highly variable. Therefore, two measurement campaigns (one in 2012 and one in 2016) with only one composite sample for each elevation step are not enough. For a correct analysis, at least 3-5 positions around the reservoir should have been sampled.

Further, sample preparation procedure for aggregate stability measurement described in 2.4.1 seems to be wrong. To measure the soil aggregate stability, it is not meaningful to remove the soil organic matter and the calcium carbonate before the measurements, as they are important factors for soil aggregate stability.  

In addition, further effects on soil aggregation (e.g. root growth, weather conditions) must be considered in the analysis.  

Reviewer 4 Report

 Impacts of water level fluctuations on soil aggregate 2 stability in the Three Gorges Reservoir, China

Abstract clear but not give any idea about lithology nor the granulometry

Keywords have to integrate riparian zone or lakes or dams, …

Introduction is clear and accurate, rich incitation but authors have to see this paper which give information about size of particles, SOC and degradation (Assessment of the effect of the use of manure and 1 lignite in sandy and clay soils

2 of water retention capacity and soil organic carbon content and its

3 mineralization in semi-arid climate) which will be published in the Arabian journal

At line 86, authors talk for the first time on the riparian ecosystem but we haven’t any idea about it and his ecological composition

At no moment, the authors consider that the phenomena is homogenous around the dam but the influence of climatic parameters and particularly the wind (wavelet) have a big influence and I think that the phenomena isn’t the same

Authors have to consider the spatial parameters in their study

Nothing on the climate parameters which could participate actively in these phenomena

Study area accurate presentation but give at the end few indications on the ecosystem which is disturbed by these fluctuations

In the sampling part, nothing on the interval sampling (5 m) : description of context show that all samples was taken in zone covered by grassland and the same soil type. What about other characteristics

Methdology correct and accurate. The only problem is the number of sample for statistical study (n=6)

As result, the SOM show her high concentration at the extreme level. Authors give an explanation related to the duration of inundation but nothing on the chemical exchange at the interface air-water-soil, on the pressure of water on the soil, …

Table2 show a correlation between size and SOM and that the high size save negatives correlation. Other studies argue the opposite

Page 235-245 other authors’ shows a high correlation with sand and less with clays. I invite the authors to look for in literature what about this correlation

Figure 3 size variability shows that the influence of dynamic of watershed is so important particularly the solid transport and the material

Nothing on the diagenesis, compaction and the sedimentary dynamic of the sites

Line 312 and 315 add “al” to the authors

In line 362, authors introduce a logical parameter which play a role in this dynamic (hydrology). Authors have to give more detail on the sediment charge of every inundation

How is the role of wave in the site study